# Comparative Transcriptomic Analysis Revealed Potential Differential Mechanisms of Grass Carp Reovirus Pathogenicity

**DOI:** 10.3390/ijms242115501

**Published:** 2023-10-24

**Authors:** Weiguang Kong, Guangyi Ding, Peng Yang, Yuqing Li, Gaofeng Cheng, Chang Cai, Jun Xiao, Hao Feng, Zhen Xu

**Affiliations:** 1Key Laboratory of Breeding Biotechnology and Sustainable Aquaculture, Institute of Hydrobiology, Chinese Academy of Sciences, Wuhan 430072, China; kongweiguang@ihb.ac.cn (W.K.); 202071741@yangtzeu.edu.cn (G.D.); youngp929@163.com (P.Y.); yuqing202012@163.com (Y.L.); chenggaofeng@ihb.ac.cn (G.C.); caichang22@mails.ucas.ac.cn (C.C.); 2State Key Laboratory of Developmental Biology of Freshwater Fish, College of Life Science, Hunan Normal University, Changsha 410081, China; xiaojun2018@hunnu.edu.cn (J.X.); fenghao@hunnu.edu.cn (H.F.)

**Keywords:** GCRV, CIK cells, RNA-seq, immune responses, metabolism

## Abstract

Grass carp reovirus (GCRV), one of the most serious pathogens threatening grass carp (*Ctenopharyngodon idella*), can lead to grass carp hemorrhagic disease (GCHD). Currently, GCRV can be divided into three genotypes, but the comparison of their pathogenic mechanisms and the host responses remain unclear. In this study, we utilized the *Ctenopharyngodon idella* kidney (CIK) model infected with GCRV to conduct comparative studies on the three genotypes. We observed a cytopathic effect (CPE) in the GCRV-I and GCRV-III groups, whereas the GCRV-II group did not show any CPE. Moreover, a consistent trend in the mRNA expression levels of antiviral-related genes across all experimental groups of CIK cells was detected via qPCR and further explored through RNA-seq analysis. Importantly, GO/KEGG enrichment analysis showed that GCRV-I, -II, and -III could all activate the immune response in CIK cells, but GCRV-II induced more intense immune responses. Intriguingly, transcriptomic analysis revealed a widespread down-regulation of metabolism processes such as steroid biosynthesis, butanoate metabolism, and N-Glycan biosynthesis in infected CIK cells. Overall, our results reveal the CIK cells showed unique responses in immunity and metabolism in the three genotypes of GCRV infection. These results provide a theoretical basis for understanding the pathogenesis and prevention and control methods of GCRV.

## 1. Introduction

Grass carp (*Ctenopharyngodon idella*) reovirus (GCRV), a member of the Reoviridae family, is responsible for causing severe grass carp hemorrhagic disease (GCHD), resulting in high mortality rates, especially in fingerling and yearling populations [1,2,3]. GCHD causes significant economic losses in the grass carp aquaculture industry in China, estimated to be at least RMB 1 billion annually based on incomplete statistics [4]. Currently, over 40 GCRV strains have been identified and classified into three main types: GCRV-I (GCRV-873), -II (GCRV-HZ08), and -III (GCRV-104) [5]. These different isolates show notable variations in terms of cytoplasmic pathogenesis, virulence, genomic electrophoresis band type, genomic sequence, and other characteristics [6,7,8,9]. Despite advancements in GCRV vaccines and drugs, it is important to recognize that these interventions may not be universally effective against all GCRV genotypes [10,11,12]. Therefore, further research is needed to understand the defense mechanism of grass carp against different types of GCRV infection.

Numerous studies have investigated virus-induced immune responses in grass carp, ranging from genomic to transcriptomic levels [13,14,15,16]. During GCRV infection, teleost fish rely on pattern recognition receptors (PRRs), primarily Toll-like receptors (TLRs) and RIG-I-like receptors (RLRs), to identify viral molecular patterns (MAMPs) or viruses [17,18]. TLRs and RLRs are crucial in initiating the immune response against viral invasion [19]. Innate immune cells phagocytose virus particles and present antigens, releasing cytokines such as chemokines, interferons, and complement factors to trigger immune responses [20]. Indeed, research findings have demonstrated contrasting immune responses in grass carp that are susceptible versus resistant to GCRV infection. GCRV-resistant grass carp can activate more immune-related pathways, including antigen processing and presentation, Toll-like receptor signaling pathway, natural killer cell-mediated cytotoxicity, and Hippo signaling pathway [15]. Transcriptomic analysis of grass carp infected with GCRV at different ages showed that the immune responses were activated more rapidly in three-year-old individuals to counteract the virus invasion [21]. To date, there are few comparative investigations of different genotypes of GCRV at the cellular level, with the majority of research focusing on GCRV-I.

An increasing number of studies has established a significant correlation between viral invasion and host metabolism [22]. Viruses heavily rely on the metabolic resources of host cells to facilitate their replication and survival, thereby inducing alterations in cellular metabolic pathways to meet their specific requirements. For instance, during the infectious spleen and kidney necrosis virus (ISKNV) infection cycle, both glucose metabolism and glutamine metabolism undergo discernible changes, with a metabolic shift observed from glucose to glutamine [23]. Furthermore, certain viruses possess the ability to hinder cellular biosynthesis in order to allocate more energy and resources towards their own synthesis. This is exemplified by viruses modifying mitochondrial metabolism to meet the energy demands necessary for their replication and proliferation within host cells [24]. In the case of grass carp, invasion by GCRV results in metabolic disorders through the manipulation of host biosynthetic and metabolic systems, which subsequently support the virus’s replication and pathogenesis [25]. However, the precise mechanisms underlying the invasion of different GCRV strains into the host remain unclear. Therefore, the aim of this study is to comprehensively investigate the immune and metabolic responses of *Ctenopharyngodon idella* kidney (CIK) cells to different types of GCRV, allowing for the identification of both divergent and convergent responses. Ultimately, these findings will provide valuable insights for the prevention and control of GCRV infection, and facilitate the breeding of GCRV-resistant grass carp.

## 2. Results

### 2.1. GCRV Infection Triggered Immune Responses in CIK Cells

In previous studies, all three genotypes of GCRV were able to infect and proliferate in CIK cells [26,27,28]. Here, we further analyzed the immune response in CIK cells with infection by three GCRVs. Compared with the control, antiviral-related genes IRF3, IRF7, NF-κB1, and IFN1 were up-regulated in CIK cells infected with GCRV-I, GCRV-II, and GCRV-III, separately (Figure 1A). Briefly, the expression levels of these genes were rapidly up-regulated at 6 h after virus infection and remained at high levels until 24 h. Moreover, CIK cells exhibited varying degrees of CPEs after infection with GCRVs (Figure 1B). For instance, CIK cells infected with GCRV-I showed significant CPE at 6 h, while the GCRV-III-infected groups showed CPE at 24 h. In contrast, the GCRV-II-infected groups did not show significant CPE from beginning to end. Referring to the infection of CIK in our studies, the virus-induced response was most significantly different at 6 and 24 h. Therefore, we selected infected groups at 6 and 24 h and control groups for further RNA-seq analysis.

### 2.2. Identification and Analysis of Differentially Expressed Genes

To explore the dynamic changes in the expression of genes, we performed differential analysis for each group. Under the stringent filtering conditions (|log_2_FC| > 2 and *p*-value < 0.05), 5331 DEGs were identified. Firstly, we found that the gene abundance in each group was homogeneous (Figure 2A), with higher abundance in the GCRV groups than in the control groups, suggesting that many genes were up-regulated in the CIK cells after GCRV infection. Additionally, the Upset plot showed that there were 6081, 163, and 396 DEGs independently expressed in GV_I_6h, GV_II_6h, and GV_III_6h groups, respectively, with 61 DEGs expressed in all 3 groups (Figure 2B). Similarly, there were 465, 632, and 91 DEGs independently expressed in GV_I_24h, GV_II_24h, and GV_III_24h groups, respectively, with 63 DEGs expressed in all 3 groups (Figure 2C). Moreover, 894 (429 up-regulated and 465 down-regulated) and 670 (438 up-regulated and 232 down-regulated) DEGs were found in the Con_6 h vs. GV_I_6h, and Con_24h vs. GV_I_24h comparison groups (Figure 2D), respectively; 1624 (559 up- and 1065 down-regulated) and 1003 (527 up- and 476 down-regulated) DEGs were found in the Con_6 h vs. GV_II_6h, and Con_24h vs. GV_II_24h comparison groups (Figure 2E), respectively; 765 (323 up- and 442 down-regulated) and 378 (265 up- and 113 down-regulated) DEGs were found in the Con_6 h vs. GV_III_6h, and Con_24h vs. GV_III_24h comparison groups (Figure 2F), respectively. Interestingly, the number of down-regulated genes in the GV_II group was significantly higher than the other two groups, and all groups generally had a higher number of down-regulated genes at 6 h than at 24 h, showing the same trend. Combined with the above results, we speculated that all groups share some functional genes in the up-regulated genes.

### 2.3. Analysis of Transcriptomic Changes in the CIK Cells after GCRV Infection

To further study the transcriptomic changes in the CIK cells after GCRV infection, cluster analysis was conducted on samples at 6 and 24 h (Figure 3A,B). Additionally, to examine congruency among biological replicates, principal component analysis (PCA) was performed on the filtered genes and visualized the groups Con, GV_I, GV_II, and GV_III at 6 and 24 h (Figure 3C,D). The PCA plot showed that the groups of the Con, GV_I, GV_II, and GV_III were separated from each other, with a high level of consistency present in the biological replicates of the same sites of tissues. Furthermore, GO enrichment analysis indicated the top two enriched processes in the categories of “biological process”, “cellular component”, and “molecular function” (biological process: cellular process and biological regulation; cellular component: cell part and organelle; molecular function: binding and catalytic activity) (Figure 4A,B). Accordingly, the GO/KEGG enrichment analyses of the biological processes and signaling pathways in different infected groups indicated that the DEGs of GV_I groups were predominantly associated with mitochondrial protein-containing complex, calcium ion binding, cell adhesion, prion disease, thermogenesis, and focal adhesion. The DEGs in GV_II were predominantly associated with lipid metabolic process, external encapsulating structure, extracellular matrix, autophagy—animal, JAK-STAT signaling pathway, PI3K-Akt signaling pathway, and protein digestion and absorption. The DEGs in GV_III were predominantly associated with the structural constituent of chromatin, signaling receptor binding, extracellular space, systemic lupus erythematosus, the TNF signaling pathway, and the relaxin signaling pathway (Figure 5A,B).

### 2.4. Immune-Related Gene Analysis in CIK Cells Infected with GCRV

GCRV infection can affect the normal physiological functions of host cells, cause cell damage, and induce robust immune responses, which have been fully confirmed in the past [29,30,31]. As shown in Figure 5, different subtypes of GCRV generated independent immune responses at different time points, and cluster analysis further showed that all immune-related genes in DEGs, including Con, GV_I, GV_II, and GV_III groups at 6 and 24 h (Figure 6A). Here, the overlap of up-regulated and down-regulated immune-related DEGs between all groups was displayed through the Upset plot (Figure 6B,C). Additionally, we found that at 6 and 24 h, there were a large number of shared genes in the up-regulated immune-related genes of GV_II and GV_III groups, while among the down-regulated immune-related genes, GV_II groups had the most independently expressed genes. To further analyze immune-related genes, the GO enrichment analyses (Figure 7A) of the biological processes in different infected groups indicated that the “immune system process” was significantly enriched in all groups, with “leukocyte chemotaxis” and “myeloid leukocyte migration” enriched in GV_II_6h, the “innate immune response” and “cytokine receptor binding” enriched in GV_III_6h, and the “regulation of immune system process” and “glucose dehydrogenase activity” enriched in GV_I_24h, “receptor ligand activity” and “signaling receiver activator activity” enriched in GV_II_24h, while “granulocyte migration” and “neutral migration” were enriched in GV_III_24h. Moreover, the KEGG enrichment analyses (Figure 7B) of the signaling pathways in different infected groups indicated that the “Toll-like receptor” and “cytokine–cytokine receptor interaction” were significantly enriched in all groups, “legionellosis” was enriched in GV_I_6h, “autoimmune thyroid release” was enriched in GV_III_6h, “type I diabetes memories” were enriched in GV_I_24h, and “NF kappa B” was enriched in GV_III_24h. Furthermore, the GO/KEGG enrichment analyses of the biological processes and signaling pathways in up-regulated immune-related DEGs indicated that “immune system process” was generally enriched in CIK cells samples infected with viruses (Figure 8A), while immune-related signaling pathways such as “viral protein interaction with cytotoxic and cytotoxic receptor”, “Toll-like receptor”, and “chemokine” were also enriched in all samples (Figure 8B). Importantly, all samples shared partial up-regulated immune-related DEGs that were enriched in biological processes and signaling pathways, including response to stimulus, immune response, immune system process, inflammatory response, defense response, response to stress, Toll-like receptor signaling pathway, and coronavirus disease COVID-19. Collectively, our results indicated that three different subtypes of GCRV have similar immune responses, and we further speculated that there may be relatively consistent targets for virus–cell interactions.

### 2.5. Metabolism-Related Gene Analysis in CIK Cells Infected with GCRV

Notably, as seen in Figure 5, we also observed that differentially expressed genes (DEGs) exhibited enrichment in metabolism-related processes and pathways at distinct time points following various GCRV infections. For instance, DEGs in all groups are enriched in sterol biosynthetic process, cholesterol metabolic process, secondary alcohol biosynthetic process, cholesterol biosynthetic process, steroid biosynthetic process, secondary alcohol metabolic process, steroid metabolic process, steroid biosynthesis, and terpenoid backbone biosynthesis. Furthermore, the gene set enrichment analysis (GSEA) showed a significant decline in immune response-related biological processes, specifically steroid biosynthesis and butanoate metabolism, at both 6 h and 24 h after infection with three distinct types of GCRV (Figure 9A,B). Notably, among different infection groups, the enrichment of biological processes associated with N-Glycan biosynthesis was observed at 6 h, while these DEGs were down-regulated at 24 h (Appendix A). This observation suggested that N-Glycan biosynthesis may effectively participate in the early stages of the antiviral response. However, in the later stages of the virus infection, the virus hijacked this process, rendering it unable to continue its primary function.

### 2.6. Validation of RNA-Seq Results by qPCR

To validate the DEGs identified through RNA-seq analysis, ten specific genes were selected for further evaluation via qPCR analysis, including up-regulated and down-regulated DEGs. The qPCR results showed significant and identical expression trends to those of the RNA sequencing data (Figure 10A–C). Therefore, the qPCR analysis results effectively validated the expressions of the DEGs identified through high-throughput sequencing analysis.

## 3. Discussion

In recent years, significant strides have been made in the identification of genomic sequences for various strains of grass carp reovirus (GCRV), such as GCRV-I, -II, and -III [32,33,34]. Remarkably, both GCRV-I and GCRV-III have shown their ability to infect CIK cells and induce significant cytopathic effects (CPE). Conversely, GCRV-II has been linked to a high fish mortality rate of 80%, despite the absence of observable CPE in CIK cells. Although studies have examined differences in nucleotide sequences, virus-encoded protein structures, and pathogenicity among different GCRV types in grass carp and CIK cells [35,36], the defense mechanisms of CIK cells against distinct GCRV types remain largely unexplored. Thus, further investigation is warranted to elucidate the divergences and commonalities in the defense mechanisms of CIK cells against three types of GCRV.

In previous studies, CIK cells have been identified as susceptible cell lines for three strains of GCRV [26,27,28,29]. We investigated the expression levels of antiviral-related genes at various time points during infection with these three GCRV strains. Our findings further support the use of CIK cells as an in vitro infection model for studying virus-induced responses to all three strains. At 6 h, GCRV-I exhibited significant cytopathic effects (CPE), while noticeable CPE was observed for GCRV-III at 24 h. This aligns with previous reports suggesting that GCRV-I has a stronger cellular infectivity compared to GCRV-III. As expected, GCRV-II did not induce significant CPE, which might be attributed to the absence of the fusion-associated small transmembrane (FAST) protein in its genome [37,38].

To investigate the immune responses triggered by the three strains, we conducted transcriptome sequencing at 6 and 24 h. The transcriptomic analysis revealed that at 6 h, GCRV-II induced the highest number of differentially expressed genes (1621), while GCRV-I and GCRV-III induced fewer differentially expressed genes, with 894 and 764, respectively. At 24 h, the overall number of differentially expressed genes decreased, but the order remained the same: GCRV-II > GCRV-I > GCRV-III. These findings are consistent with previous in vivo transcriptomic studies, where GCRV-II induced a higher number of differentially expressed genes compared to GCRV-I [21]. We further assessed the distribution of up-regulated and down-regulated genes and observed that GCRV-II down-regulated a significantly larger number of host genes, almost double the number induced by other GCRV strains. These results highlight noteworthy distinctions in the response patterns induced by the three GCRV strains, and additional analysis of their respective response functions is currently underway.

Initially, we conducted a thorough analysis of all differentially expressed genes following GCRV infection. GO analysis uncovered that GCRV infection predominantly impacts cellular processes, biological regulation, and immune system processes, as well as cell parts, organelles, and membranes. Moreover, GCRV can influence binding, catalytic activity, and molecular function regulators. In accordance with KEGG enrichment analyses, GCRV infection triggered the activation of numerous cellular immune pathways. At 6 h, GCRV-II notably activated the JAK-STAT signaling pathway, autophagy, cytokine–cytokine receptor signaling pathway, and Toll-like signaling pathway. These pathways are known to play crucial roles in antiviral defense [39,40,41,42].

In comparison to GCRV-II, GCRV-III exhibited a lower level of activation in certain immune signaling pathways, while also being capable of activating pathways such as IL-17 and systemic lupus erythematosus, suggesting the induction of type II hypersensitivity reactions [43]. These reactions, also known as cytotoxic hypersensitivity reactions, are pathological immune responses characterized by cell lysis or tissue damage. They are triggered by the binding of antibodies (IgG or IgM) to antigens on the surfaces of target cells, including phagocytic cells and NK cells. It is noteworthy that the pathways induced by GCRV-I are primarily associated with diseases such as Prion disease, Huntington disease, and non-alcoholic fatty liver disease. At 24 h, GCRV-I and GCRV-II exhibit significant enrichment in the PI3K-Akt signaling pathway, focal adhesion pathways, TNF signaling pathway, and viral protein interaction with cytokine and cytokine receptor pathways, which are known to be essential for the antiviral responses [44]. The downstream targets of the PI3K/Akt pathway include mammalian target of rapamycin (mTOR) [45]. GCRV induces autophagy, which enhances virus replication through the Akt/mTOR pathway [40]. These data further suggest that the immune response of CIK cells differs depending on the type of GCRV.

To enhance our understanding of the immune response processes elicited by the three types of GCRV, we carried out additional analysis on immune-related genes. In our findings, we observed that at 6 h, GCRV-I demonstrated a higher up-regulation of immune genes compared to the other two strains. Conversely, at 24 h, GCRV-II induced a greater up-regulation of immune genes. Similarly, at the individual level, GCRV-I and -II led to more up-regulation of immune-related genes on the first day and second day, respectively [16]. These results align with the virulence and latency period characteristics of the two viruses. Furthermore, in comparison to GCRV-I, GCRV-II exhibited activation of processes such as G protein-coupled receptor binding and extracellular space, whereas GCRV-III triggered responses such as the inflammatory response and cytokine receptor binding. KEGG analysis further revealed that at 24 h, GCRV-I exhibited almost negligible activation of the Chemokine signaling pathway, while GCRV-II induced higher levels of chemokines and other immune pathways. This heightened immune response could potentially serve as a double-edged sword for the host. While it enhances virus clearance at the cellular level, it may also exacerbate pathological processes at the individual level [46]. Further investigation is required to establish a direct relationship between the host’s immune response and the virus. The aforementioned results suggest that the three strains induce distinct immune responses in the host, necessitating different intervention measures for each strain. Additionally, our analysis of the shared immune responses to the three strains reveals the Toll-Like signaling pathway’s positive role in immune responses against all three viruses. In teleosts, TLR3 and TLR19 recognize viral dsRNA and activate the Ikkε/Ikki/TBK1 complex, resulting in the expression of type I interferons [47]. Other TLR genes, such as TLR22a, TLR22b, TLR7, and TLR8, also exhibit responsiveness to GCRV and display up-regulation. These findings hold potential for the development of molecular markers in molecular breeding.

Interestingly, our transcriptomic analysis aligns with previous findings that viral infection disrupts host cell metabolism [22]. At 6 h, all three GCRV strains exhibited a significant impact on lipid synthesis metabolic pathways, including steroid, cholesterol, sterol, and secondary alcohol biosynthetic processes. Viral infection promoted the expression of IDH1/2, which facilitates viral replication, and lipid synthesis is crucial for the maturation of viral particles [48]. Notably, GCRV-II displayed an additional abundance of differentially expressed genes involved in lipid biosynthesis and metabolic processes [49]. Ultrastructural analysis of GCRV-II-infected CIK cells revealed the presence of an extra lipid membrane surrounding viral particles [50]. Since GCRV is a non-enveloped virus, we speculate that this lipid may facilitate viral particle proliferation and maturation, but further research is needed to confirm this. Additionally, we found that GCRV-I specifically activated processes related to mitochondrial protein-containing complex, including inner mitochondrial membrane protein complex and oxidoreductase complex. This suggested GCRV-I may affect the host’s mitochondria, altering energy metabolism to promote viral reproduction. Further KEGG analysis revealed that at 6 h post infection, GCRV-I-induced genes primarily contribute to oxidative phosphorylation, thermogenesis, and non-alcoholic fatty liver disease. GCRV-I potentially exploits the energy metabolism of CIK cells, with the subsequent increase in reactive oxygen species production acting as a critical factor leading to cell death. Moreover, we observed a down-regulation of specific metabolic pathways across all three viral groups, namely, steroid biosynthesis, butanoate metabolism, and N-Glycan biosynthesis. These pathways are closely associated with antiviral function, warranting further investigation into the mechanisms by which the three GCRV strains inhibit these pathways.

## 4. Materials and Methods

### 4.1. Virus, Cells, and Infection

GCRV strains were generously provided by Prof. Yaping Wang of Institute of Hydrobiology, Chinese Academy of Science. The CIK line was provided by the China Center for Type Culture Collection (CCTCC). CIK cells were incubated at 28 °C and maintained in low-glucose Dulbecco’s modified Eagle’s medium (DMEM, Gibco, Gaithersburg, MD, USA) supplemented with 10% fetal bovine serum and 1% (*v*/*v*) penicillin–streptomycin in a humidified atmosphere with 5% CO_2_. For virus infection in cells, CIK cells were plated in 96-well plates for 24 h in advance and then infected with GCRV. After 3 h, the virus inoculum was removed and the cells were incubated with a new medium. The control groups were treated with PBS. Then, when CPEs were clearly apparent, images of infected cells were captured using a microscope.

### 4.2. RNA Isolation and Quantitative Real-Time PCR (qPCR) Analysis

Total RNA was extracted from various tissues using a Trizol Reagent (Invitrogen, Carlsbad, CA, USA) according to the manufacturer’s protocol. The concentration of extracted RNA was determined by spectrophotometry (Nanodrop ND2000, Thermo Fisher Scientific, Wilmington, DE, USA), and the integrity of the RNA was determined by 1% agarose gel electrophoresis (Agilent Bioanalyser, Santa Clara, CA, USA). To normalize gene expression levels for each sample, equivalent amounts of total RNA (1000 ng) were used for cDNA synthesis with the SuperScript first-strand synthesis system in a 20 μL reaction volume. The synthesized cDNA was diluted three times. qPCR was performed to detect the expression levels of immune-related genes. All primers used in the study were designed by software Primer Premier 5.0 and are listed in Table 1. The relative expression ratio of the selected gene vs. elongation factor 1α (EF1α) (reference gene) was calculated using the 2^−∆∆Ct^ method. Reactions of SYBR Green were performed in a 10 µL volume containing 5 µL of 2 × SYBR Green qPCR Mix (YEASEN, Shanghai, China), 1 µL of each forward and reverse primer (10 µM), 3 µL of water, and 1 µL of diluted cDNA (50 ng/µL). All experiments were performed in triplicate.

### 4.3. RNA Isolation, Library Construction, and Sequencing

CIK cells were sampled at 0, 6 and 24 h after GCRV infection (named Con_6h, Con_24h, GV_1_6h, GV_2_6h, GV_3_6h, GV_1_24h, GV_2_24h, and GV_3_24h; three biological duplicates for each group), and sent to Majorbio Technology Co., Ltd. (Wuhan, China). Briefly, total RNA was extracted with Trizol reagent (Invitrogen, Carlsbad, CA, USA). Sequencing libraries were generated using NEBNext^®^ UltraTM RNA Library Prep Kit for Illumina^®^ (NEB, Ipswich, MA, USA) following the manufacturer’s recommendations. Briefly, mRNA was purified from total RNA using poly-T oligo-attached magnetic beads. Fragmentation was carried out using divalent cations under elevated temperature in NEBNext First-Strand Synthesis Reaction Buffer. First-strand cDNA was synthesized using random hexamer primers. Second-strand cDNA synthesis was subsequently performed using DNA Polymerase I and RNase H. After adenylation of 3′ ends of DNA fragments, NEBNext adaptors with hairpin-loop structure were ligated to prepare for hybridization. In order to select cDNA fragments of preferentially 250–300 bp in length, the library fragments were purified with the AMPure XP system (Beckman Coulter, Brea, CA, USA). Then, 3 µL USER Enzyme (NEB, Ipswich, MA, USA) was used with size-selected, adaptor-ligated cDNA at 37 °C for 15 min, followed by 5 min at 95 °C before PCR. Then, PCR was performed with Phusion High-Fidelity DNA polymerase, Universal PCR primers, and Index (X) primer. At last, PCR products were purified (AMPure XP system), and library quality was assessed. The library preparations were sequenced on an Illumina Novaseq platform, and 150 bp paired-end reads were generated.

### 4.4. Differential Expression Analysis of Transcriptome Sequencing and Validation by qPCR

The grass carp genome (accession number: PRJEB5920) was used as a reference genome for further analysis. DESeq2 R package (1.16.1) was used to perform differential expression analysis [51]. Here, 24 samples from the two time points mentioned above were divided into 8 groups (Con6h, Con24h, GVI6h, GVI24h, GVII6h, GVII24h, GVIII6h, and GVIII24h) for transcriptome sequencing. A total of 1,144,052,096 high-quality clean data were obtained after a series of quality controls, and the percentage of Q30 base in all samples was no less than 94.05% (Appendix A). The resulting *p*-values were adjusted using Benjamini and Hochberg’s approach for controlling the false discovery rate. Genes with an adjusted *p*-value < 0.05, found by DESeq2, were assigned as differentially expressed genes (DEGs).

Gene ontology (GO) classification, comprising GO-BP (biological process), GO-MF (molecular function), and GO-CC (cellular component), implemented by the cluster Profiler R package (3.12.1) was applied to uncover the functions of intersecting genes [52,53]. GO terms with a corrected *p*-value < 0.05 were considered significantly enriched by DEGs. The Kyoto Encyclopedia of Genes and Genomes (KEGG) database is used for understanding high-level functional information in biological systems from molecules, cells, organisms, and ecosystems, and it is particularly powerful for large-scale molecular data sets generated by genome sequencing and other high-throughput experimental approaches [54]. Cluster Profiler R package was used to test the statistical enrichment of DEGs in KEGG pathways [52,53]. KEGG terms with corrected *p*-values of less than 0.05 were considered significant. Gene set enrichment analysis (GSEA) is a computational approach to determine if a predefined gene set can show a significant, consistent difference between two biological states. GO and KEGG data sets were used for GSEA independently.

qPCR was performed to validate the DEGs identified from transcriptome sequencing. Ten DEGs were randomly selected for qPCR validation. The detection was performed in triplicate for each biological replicate. The relative expression values of selected genes were calculated using the 2^−ΔΔCt^ method and normalized against the expression levels of the EF1α gene.

### 4.5. Statistical Analysis

The statistical results were expressed as mean ± standard error estimate. The diagrams were created using GraphPad 5.0. The results were obtained from three independent experiments and each was performed in triplicate.

## 5. Conclusions

In conclusion, our transcriptomic analysis revealed the impact of GCRV infection on CIK cells. We observed that GCRV primarily affects the immune response and metabolism of CIK cells. In terms of the immune response, we identified alterations in immune-related genes, including cytokines, inflammatory factors, and immune regulatory factors. These changes suggest that GCRV either activates or inhibits the immune response in CIK cells, potentially influencing the body’s ability to resist infection. In relation to metabolism, we noted disruptions in various metabolic pathways such as lipid metabolism, sterol metabolism, and cholesterol metabolism upon GCRV infection. These disruptions can affect the energy supply and biosynthesis processes of CIK cells, ultimately impacting their normal functions. Taken together, these findings significantly contribute to our understanding of the infection mechanism of GCRV and the development of prevention and treatment strategies. Further studies are warranted to elucidate the detailed mechanisms underlying the interaction between GCRV infection and CIK cells, providing a theoretical foundation for the prevention and control of GCRV infection in grass carp.

## Figures and Tables

**Figure 1 ijms-24-15501-f001:**
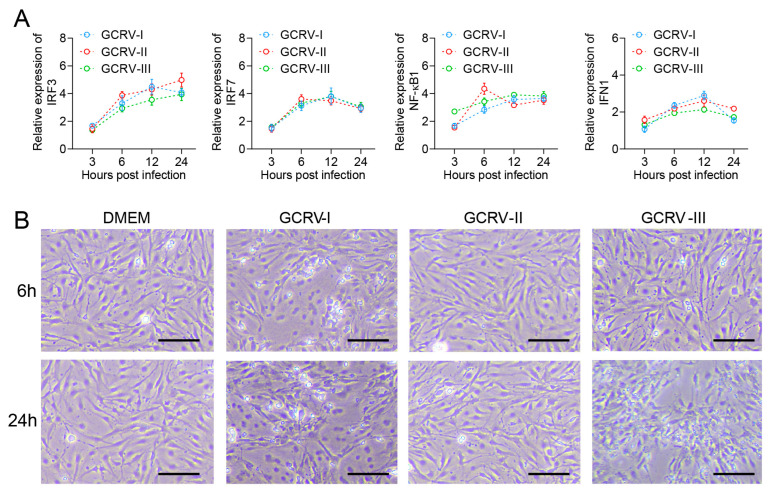
GCRV infection triggered expressions of immune genes and CPEs in CIK cells. (**A**) The relative expression levels of IRF3, IRF7, NF-κB1, and IFN1 genes in CIK cells after being infected with three subtypes of GCRV at 3, 6, 12, and 24 h. Data are representative of three independent experiments (mean ± SEM) (*n* = 6). (**B**) CPEs of CIK cells after infection with GCRVs at 6 and 24 h. In electron microscopy (EM) analysis, the scale bar is 100 μm.

**Figure 2 ijms-24-15501-f002:**
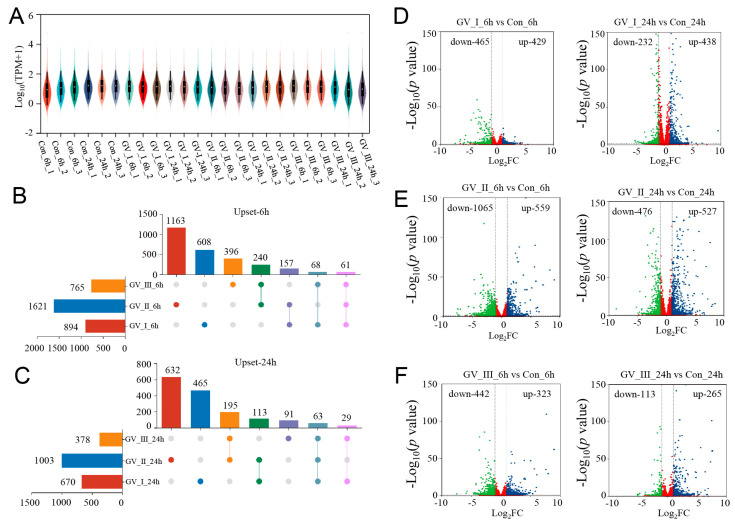
DEG analysis of transcriptome sequencing after GCRV infection. (**A**) Violin plot of expression abundance for each sample after GCRV infected CIK cells. The white dots represent the median; the black rectangles represent the range from the lower quartile to the upper quartile; the outer shapes of the black rectangle represent an estimate of the kernel density. The length of the longitudinal axis represents the degree of dispersion. (**B**) Upset plot showing differential expression genes of overlap or unique responses in CIK cells after infection with GCRV at 6 h compared to control. (**C**) Upset plot showing differential expression genes of overlap or unique responses in CIK cells after infection with GCRV at 24 h compared to control. (**D**) Volcano plots showing the DEG distribution of CIK cells infected with GCRV-I at 6 h (left) and 24 h (right) compared to control. (**E**) Volcano plots showing the DEG distribution of CIK cells infected with GCRV-II at 6 h (left) and 24 h (right) compared to control. (**F**) Volcano plots showing the DEG distribution of CIK cells infected with GCRV-III at 6 h (left) and 24 h (right) compared to control. Blue spots, expression fold change > 2, *p*-value < 0.05. Green spots, expression fold change < 2, *p*-value < 0.05. Red spots indicate no difference. The vertical axis represents log_10_ (*p*-value), and the horizontal axis represents log_2_ (fold change).

**Figure 3 ijms-24-15501-f003:**
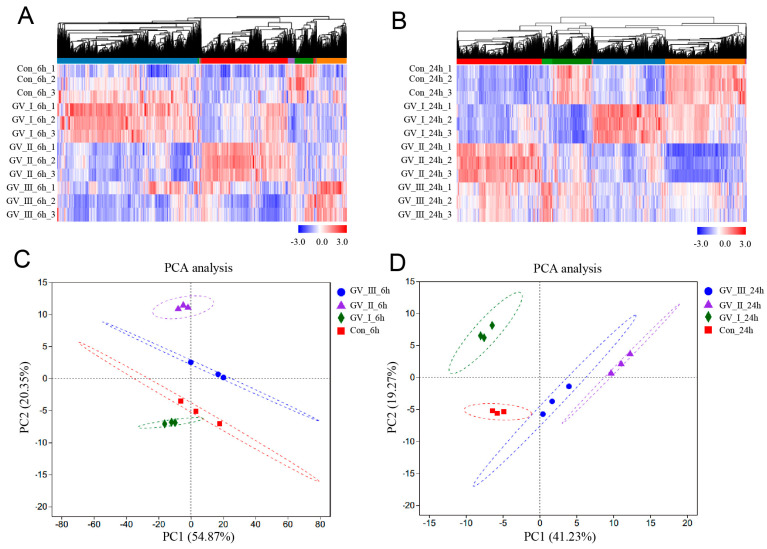
Cluster analysis and correlation analysis of DEGs in CIK cells after viral infection. (**A**) Heat map analysis of all DEGs in 6 h groups compared with the control group. (**B**) Heat map analysis of all DEGs in 24 h groups compared with the control group. The colors represent the relative abundance of the corresponding genus, with red indicating higher abundance and blue indicating lower abundance. Pearson correlation was carried out and the “Complete” method was used to cluster values. (**C**) PCA cluster plots of gene expression levels in samples at 6 h. (**D**) PCA cluster plots of gene expression levels in samples at 24 h. The horizontal axis represents the first-ranking principal component dimension, and the vertical axis represents the second-ranking principal component dimension. Different shapes and colors indicate different groups.

**Figure 4 ijms-24-15501-f004:**
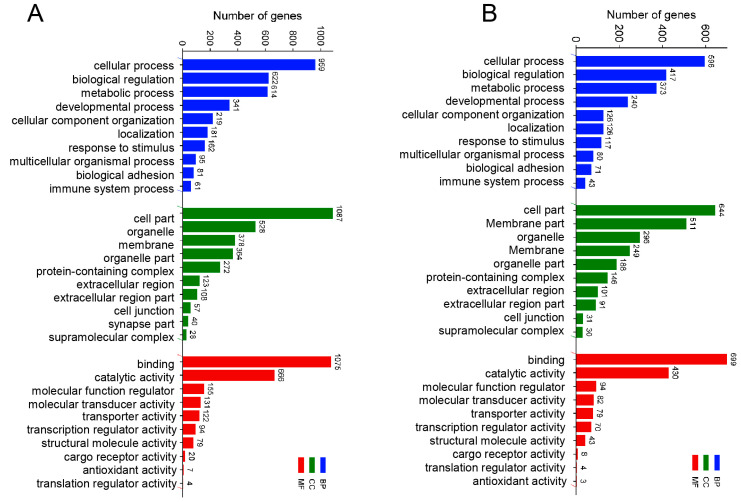
GO enrichment analysis of total DEGs in three types of GCRV infection group. (**A**) 6 h groups compared with the control groups. (**B**) 24 h groups compared with the control groups. The *y*-axis represents the gene ontology process, and the *x*-axis represents the number of genes in the process.

**Figure 5 ijms-24-15501-f005:**
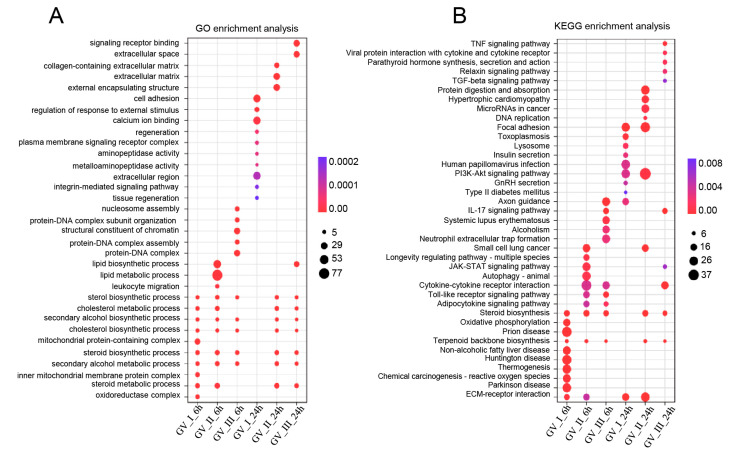
Enrichment analysis of all DEGs in CIK cells infected with GCRVs. (**A**) GO enrichment analysis of the biological processes of the DEGs in different infected groups. (**B**) KEGG enrichment analysis of the signaling pathways of the DEGs in different infected groups.

**Figure 6 ijms-24-15501-f006:**
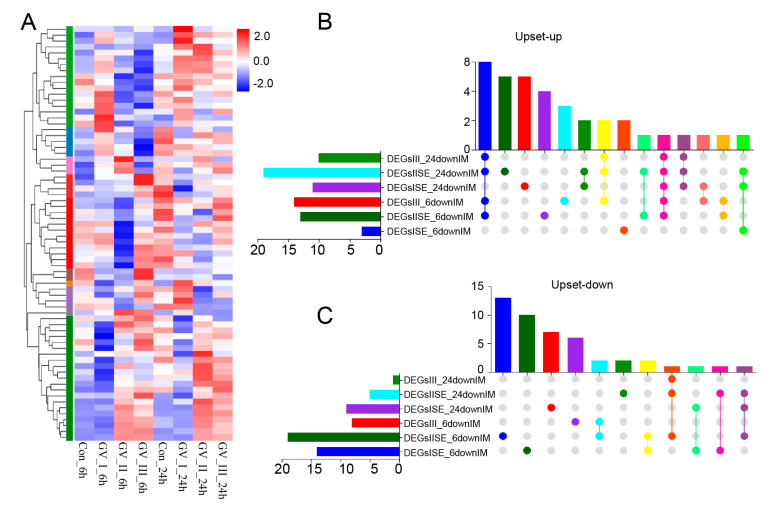
Differentially expressed immune-related gene analysis in CIK cells infected with GCRV. (**A**) Heat map and cluster analysis showing the average classification composition of the differentially expressed immune-related genes in all groups. The colors represent the relative abundance of the corresponding genus, with red indicating higher abundance and blue indicating lower abundance. Pearson correlation was carried out and the “Complete” method was used to cluster values. Upset plot showing the overlap of up-regulated (**B**) and down-regulated (**C**) immune-related genes in all groups.

**Figure 7 ijms-24-15501-f007:**
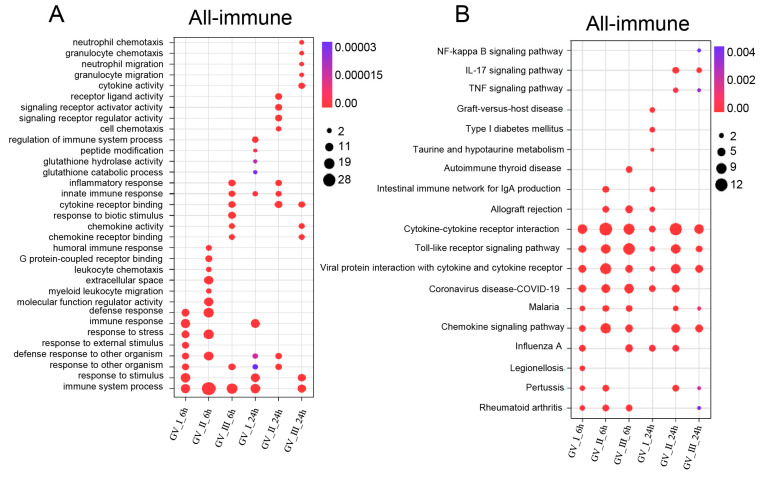
Enrichment analysis of immune-related DEGs in CIK cells infected with GCRVs. (**A**) GO enrichment analysis of the biological processes of immune-related DEGs in all groups. (**B**) KEGG enrichment analysis of the signaling pathways of immune-related DEGs in all groups.

**Figure 8 ijms-24-15501-f008:**
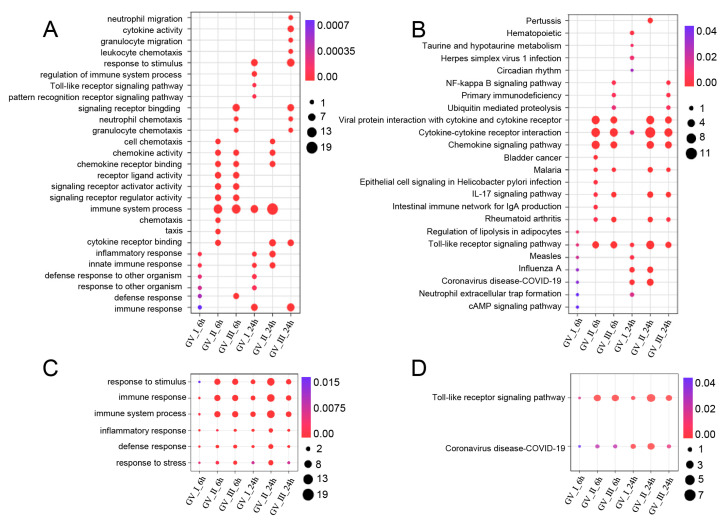
Enrichment analysis of up-regulated immune-related DEGs in CIK cells infected with GCRV. (**A**) GO enrichment analysis of the biological processes of up-regulated immune-related DEGs among different infected groups. (**B**) KEGG enrichment analysis of the signaling pathways of up-regulated immune-related DEGs among different infected groups. (**C**) Common up-regulated biological processes among different infection groups. (**D**) Common up-regulated signaling pathways among different infection groups.

**Figure 9 ijms-24-15501-f009:**
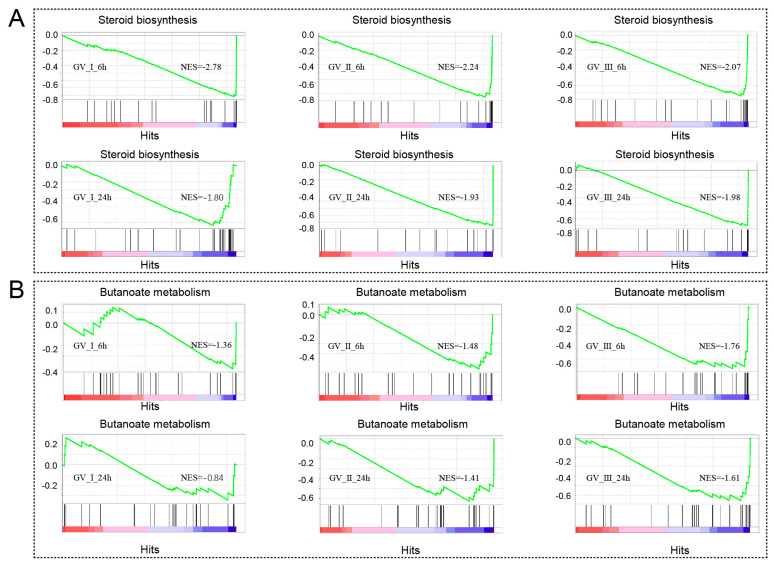
GSEA was conducted to analyze the down-regulated genes in CIK cells at 6 and 12 h with three different types of GCRV. The DEGs exhibited a down-regulation pattern, particularly within the processes related to steroid biosynthesis (**A**) and butanoate metabolism (**B**).

**Figure 10 ijms-24-15501-f010:**
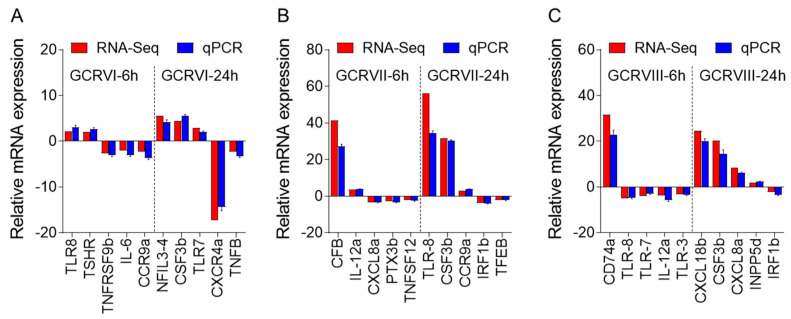
Comparison of the expression profiles of ten selected genes as determined by RNA sequencing and qPCR. (**A**) Comparison of up-regulated genes and down-regulated genes in CIK cells infected with GCRV-I at 6 and 24h. (**B**) Comparison of up-regulated genes and down-regulated genes in CIK cells infected with GCRV-II at 6 and 24 h. (**C**) Comparison of up-regulated genes and down-regulated genes in CIK cells infected with GCRV-III at 6 and 24 h. Data are representative of three independent experiments (mean ± SEM).

**Table 1 ijms-24-15501-t001:** Primers used in this study.

Gene	Primer Sequence (5′-3′)		Product Size
	Forward Primer	Reverse Primer	
EF1α	CGCCAGTGTTGCCTTCGT	CGCTCAATCTTCCATCCCTT	99
IRF3	ACTTCAGCAGTTTAGCATTCCC	GCAGCATCGTTCTTGTTGTCA	208
IRF7	CGCCTGTGTTCGTCACTCGT	GGTGGTTGGAAAGCGTATTGG	105
NF-κB1	CCAGGTGCGGTTTTATGAAGATGA	ATGGCTTGGGTTCGCTCGTTT	221
IFN1	AAGCAACGAGTCTTTGAGCCT	GCGTCCTGGAAATGACACCT	79
TLR8	CGAGTGCCCAGAGGATTACC	AGCTCGCGGATGGATTTCAA	201
TSHR	TTACCGTCCAAACCGAGTGG	CCGATCACACACACAGGTGA	121
IL-6	CAGCTCCAGGTGAGTGAAGG	GGTGTCCACCCTTCCTCTTG	172
CCR9a	ACCAACATCATCCGCACCTT	GTCGTATTGGCTGCCTGAGT	152
TNFRSF9b	CGTCTAGTCAGTCAGTGCGG	CTGGGGGTGCATTTCTCCTT	164
TLR7	CGATCAAGGTCGGTCCCAAA	GGAAGGCCTTTGGGTATGCT	136
NFIL3-4	CTCACGGATGAGGTGGGAAC	CGGCGTTTCTCCCAGTAAGT	126
CSF3b	GCTGGCACAGATTCCCAGTA	TTGTGTCGCAGGCTTGGTTA	197
CXCR4a	AGGAAACTCCTCGCCAATCG	GTCCTGAGGGTAAATGCGCT	138
TNFB	GTCTGCCAATCACACGAGGA	AACGCAAACACGCCAAAGAA	158
CFB	AGAATGGAGAGGTGACCCCA	CTGTTCGACTGCTACCAGGG	204
IL-12a	AGCCCATCGCCACTGATAAC	ATGAGGTCTTTTGTGGCCGT	175
CXCL8a	ACACCTACAGCATCGAGCAT	GAGGGCTAGGAGGGTAGAGC	248
PTX3b	ACTGCCCGGCAACATTATGA	CTCCGTACCTTAGGGGCTCT	178
TNFSF12	AACGTTTGCAACAGGGTTCG	CGGGTCGTTTACCGTTCTGA	148
IRF1b	TCGTGTGGGTCAACAAGGAG	TTTACCTGTGTGGATCGCCC	122
TFEB	GGTGTCGCACCTATGGGAAA	CCCAAGACTGTCCAGGTCAC	181
TLR3	CTTGATGCTTTGCGTGGCTT	TCTGTCAGGTTGGACAACGG	182
CD74a	GCCATGAGATTCCGGACACA	CACACGGACCGACTGATCTT	166
CXCL18b	TAAGACCACGCTGCGACAAA	TCCAGCAAGTCTGCAGGTTC	126
INPP5d	CTCTCCAAAGCAGGCAAGGA	CAGGTCTGGCAGCAAAGAGA	192

## Data Availability

The raw RNA sequencing data have been deposited in the NCBI Sequence Read Archive under BioProject accession number PRJNA1012803.

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
