# Peer review of "Comparative Transcriptomic Analysis Revealed Potential Differential Mechanisms of Grass Carp Reovirus Pathogenicity"

_ijms, 2023, doi:10.3390/ijms242115501_

Round 1

Reviewer 1 Report

The authors investigated the immune and metabolic responses of Ctenopharyngodon idella kidney cells to different types of GCRV. There are several issues that need to be amended.

 Comments:

Table S1: Add PCR product size for each gene.

Figures (2-9): Please increase the font size in all figures and improve the quality of the figures.

Figure 1b: Please prove high resolution of cell culture pictures (200X) of each treatment experiment to see the clear effect of virus on the cells.

Line 371, Add passage number of CIK cells used.

Line 420, It is not clarified that which database was used for Gene Ontology. Do you mean human?

Line 422, add version of R package.

Section 4.2. Gene expression analysis should come immediately afterwards section. 4.4. Differential expression analysis of transcriptome sequencing. Please pay attention here.

Validation of RNA-seq results is missing in methods. On what basis, ten specific genes were selected for validation.

Line 382, authors used only one reference gene (beta actin) for normalize the qRT-PCR data, however, three reference genes are needed as per the international guidelines.

Reviewer 2 Report

This study focuses on the comparative transcriptomic analysis of grass carp (Ctenopharyngodon Idella) reovirus (GCRV) three genotypes with regards to their pathogenic mechanisms and the host response in CIK cells. The authors found a cytopathic effect (CPE) of the GCRV-I and GCRV-III groups, whereas the GCRV-II group did not show any CPE. GCRV-I, II and III could all activate the immune response in CIK cells, but GCRV-II induced a more intense response.  

The study is well performed. The results are interesting, and the manuscript is well written both scientifically and the English language.

Line 15: CIK model; abbreviations should be written in full the first time they are mentioned.

L 20: could active; change to: could activate

L 57: few comparative investigating about; change to: fewcomparative investigations about

L 59: An increasing number of studies have; change to: An increasing number of studies has

L 74: Ctenopharyngodon Idella kidney; change to: Ctenopharyngodon idella kidney

The English language is fine

Reviewer 3 Report

The study examines the pathophysiological response of the grass carp by examining the transcriptomic changes after infection of a particular category of cells with reovirus strains. The study is both interesting and well structured, following a good methodological approach. I have only some minor comments that follow in the next lines and a major suggestion. My suggestion is to add a small paragraph in the discussion or maybe a graphical abstract proposing the use of the most characteristic immune related genes to be used in next studies. This way the findings will be also useful for future studies and tested after exposure to other pathogens or abiotic factors that affect the health of the grass carp

Minor comments

line 15: please explain the abbreviation CIK the first time mentioned

lines 18-19: experimental groups of carps? Please be specific

line 383: -ΔΔCt should be a superscript, please correct

Some info is missing from the section 4.2. Before explaining qPCR, RNA extraction should be mentioned. Also it would be preferable to include the primers in the main text as a Table instead of supplementary material

lines 80-82 seem to fit better in the discussion

Round 2

Reviewer 1 Report

The authors have made all the necessary edits. The manuscript now appears much better readable.